# A High-Capacity Coverless Information Hiding Based on the Lowest and Highest Image Fragments

Kurnia Anggriani [1,2] , Shu-Fen Chiou [3], Nan-I Wu [4] and Min-Shiang Hwang [1,5,*]

1 Department of Computer Science & Information Engineering, Asia University, Taichung 41354, Taiwan
2 Department of Information System, University of Bengkulu, Bengkulu 38122, Indonesia
3 Department of Information Management, National Taichung University of Science and Technology, Taichung 41354, Taiwan
4 Department of Digital Multimedia, Lee-Ming Institute of Technology, New Taipei 243086, Taiwan
5 Department of Medical Research, China Medical University Hospital, China Medical University, Taichung 404327, Taiwan
* Correspondence: mshwang@asia.edu.tw

**Abstract:** Coverless data hiding is resistant to steganalytical tool attacks because a stego image is not altered. On the other hand, one of its flaws is its limited hiding capacity. Recently, a coverless data-hiding method, known as the coverless information-hiding method based on the most significant bit of the cover image (CIHMSB), has been developed. This uses the most significant bit value in the cover image by calculating the average intensity value on the fragment and mapping it with a predefined sequence. As a result, CIHMBS is resistant to attack threats such as additive Gaussian white noise (AGWN), salt-and-pepper noise attacks, low-pass filtering attacks, and Joint Photographic Experts Group (JPEG) compression attacks. However, it only has a limited hiding capacity. This paper proposes a coverless information-hiding method based on the lowest and highest values of the fragment (CIHLHF) of the cover image. According to the experimental results, the hiding capacity of CIHLHF is twice that of CIHMSB.

**Keywords:** coverless information hiding; image data hiding; most significant bit

## 1. Introduction

Data-hiding methods in the Internet era and open access for various applications are in unavoidable and continuous demand [1–6]. This has resulted in rapid research developments in the field of data hiding, also known as steganography. According to this trend, antisteganography and steganalytical research subjects are also quickly developing.

Steganography is a technique for concealing confidential messages. To conceal a message in classical steganography, a modified medium is required. This is referred to as spatial domain if the alteration is performed directly on the pixel image value [7–13]. If the histogram value and frequency of the pixel image are changed, this is known as the transform domain [14–16]. Lastly, compressed domain steganography is when changes are performed after an image compression method had been executed [17–25].

On the other hand, steganalysis is a process for detecting images that have modifications to the embedded hidden message. Then, the secret message is extracted to identify and utilize it for further purposes. Steganalytical attacks are currently the most widely used additive Gaussian white noise (AGWN), salt-and-pepper noise attacks, low-pass filtering attacks, and JPEG compression attacks. This is a significant threat in classical steganography. The current research on steganalysis is more advanced by employing machine learning and deep learning methods [26–30].

Coverless image steganography (CIS) was introduced by Zhou et al. in 2015 [31]. The scheme of Zhou et al. [31] is based on an image mapping operation. First, an indexed image database is constructed on the basis of each image hash sequence. Then, the transformation

of secret data into a bit string and several segments is operated. Lastly, a mapping operation between a secret data segment and the hash image is developed to match the appropriate image as the stego image. The scheme of Zhou et al. [31] is robust against steganalytical tools and provides resistance to common image attacks such as rescaling, luminance change, and noise addition. However, the scheme of Zhou et al. [31] has a low embedding capacity of eight bits per carrier.

To improve the embedding capacity of the scheme of Zhou et al. [31], in 2018, Zou et al. [32] proposed a CIS method based on the average pixel value of the subimages. In this approach, the secret information is segmented according to the structure of a Chinese sentence, including subject, predicate, object, and preposition. Then, a hash sequence is generated on the basis of the average pixel value of the subimages by using a hashing algorithm. After that, a mapping operation between the segmented secret information and the hash sequence is operated. Lastly, a multilevel index structure is built to retrieve the appropriate stego images. This approach achieved higher embedding capacity than that of the scheme of Zhou et al. [31], which is 80 bits per carrier. In the same year, Zhou et al. [33] developed a CIS approach on the basis of partial-duplicate picture retrieval, which has a larger embedding capacity. This approach can embed 384 bits of secret messages per carrier.

Furthermore, instead of sending the original secret images, the sender utilizes the highest similarity image in this approach. In 2019, Luo et al. [34] introduced a novel coverless information-hiding approach based on deep learning. This scheme implemented a convolutional neural network algorithm as the high-level semantic feature extracted to retrieve real-time image data hiding. Besides achieving better robustness, this approach also has higher embedding capacity than that of the previous approach, which can hide 800 bits per carrier.

In 2021, Yang et al. [35] proposed the novel coverless information-hiding method based on the most significant bit of the cover image (CIHMSB). CIHMSB is based on image mapping operation, the same as CIS. The difference is that, instead of mapping to an indexed image dataset, CIHMSB maps the secret message with the MSB of the fragment average value. As a result, the mapping flag and unmodified cover image are sent to a receiver. CIHMSB is a simple computation method. In addition, CIHMSB performed well against steganalytical attacks. Although CIHMSB achieved higher hiding capacity than that of the previous scheme above, which is 4.096 secret bits per carrier, it is still categorized as having low hiding capacity because CIHMSB only utilizes the average pixel value of image fragments. So, CIHMSB can potentially increase the hiding capacity by utilizing fragment properties, i.e., the fragment's highest and lowest pixel values.

In order to address CIHMSB's low hiding capacity, in this paper, we propose a coverless information hiding method based on the lowest and highest values of the fragment (CIHLHF) of the cover image. It can be computed supposing an image size of $256 \times 256$, and the image fragment is $4 \times 4$, so that the hiding capacity is 8192 secret bits. Experimental results showed that the CIHLHF hiding capacity was doubled. Moreover, CIHLHF performs more securely toward attackers regarding the time cost of breaking the method. Our other contribution is constructing the table for mapping the rule of embedding and extracting procedures, which makes CIHLHF more applicable for real-word and real-time security problems.

The remainder of the work is organized as follows: Section 2 explains our proposed method, coverless information hiding of the highest lowest fragments. Then, Section 3 discusses the experimental results. Lastly, Section 4 addresses the conclusions.

## 2. Proposed Method CIHLHF

To resolve the issue of CIHMSB, we developed CIHLHF. The flowchart of CIHLHF is presented in Figure 1. Moreover, the proposed CIHLHF may be classified into embedding and extracting operations, as discussed in Sections 2.1 and 2.2, respectively.

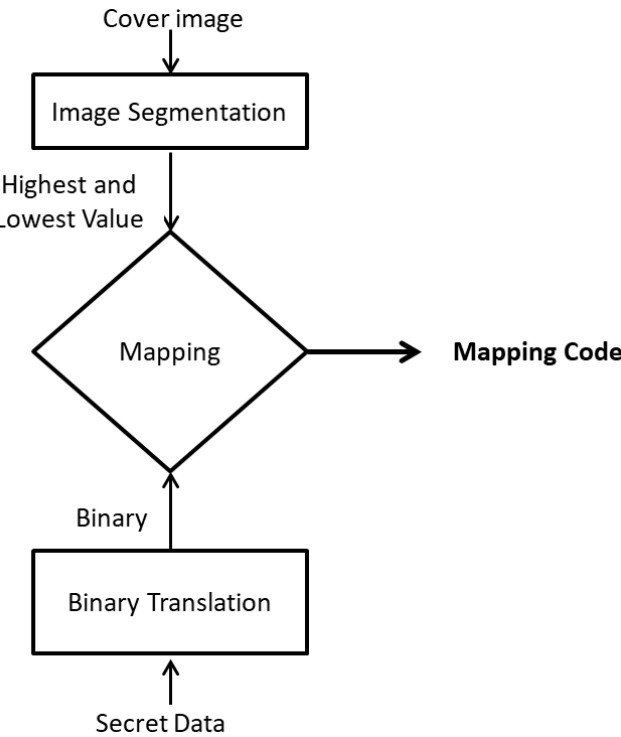

**Figure 1.** Flowchart of the proposed CIHLHF.

*2.1. Embedding Procedures*

There are three main procedures in the proposed coverless information hiding scheme: cover image preparation, secret data preparation, and mapping. First, a cover picture is converted into a segmented image $C_i$. Then, for each $C_i$, the lowest and highest, $L_i$ and $H_i$, are determined. Second, secret data are converted into binary format $M_i$. Lastly, a mapping operation is performed among the MSBs of $L_i$, $H_i$, and $M_i$.

The embedding process is explained in depth below:

Cover Image Preparation:

1. Divide the cover picture I, of size K × L pixels, into m × n nonoverlapping segments, $C_i$.
2. Determine the lowest and highest values, $L_i$ and $H_i$.

$$L_i = min(v_1C_i, v_2C_i, \ldots, v_{m \times n}C_i \\ H_i = max(v_1C_i, v_2C_i, \ldots, v_{m \times n}C_i \tag{1}$$

where $v_1C_i, \ldots v_{m \times n}C_i$ are the pixel values in the segment $C_i$.

3. Convert $L_i$ and $H_i$ into an eight-bit binary.
4. Using Equation (2), calculate the hiding capacity of the cover picture:

$$EC = \left( \frac{K \times L}{m \times n} \right) \times 2 \tag{2}$$

The total number of fragments for the cover image with K × L pixels into m × n non-overlapping segments is (K × L)/(m × n). Since the proposed hiding scheme is hidden on the 2 pixels (the lower and higher pixels), each fragment can be hidden to 2 bits.

Secret Data Preparation:

Convert the secret data $T_i$ into a seven-binary (an ASCII code) format.

For example, the ASCII code of character A is 65 in a decimal value. Therefore, the character A of secret information (T) is (1,0,0,0,0,0,1).

Mapping:

1. Determine the predetermined mapping key Z between a sender and a receiver, where the length of Z is the same as EC.
2. Create a mapping between $T_i$ and the MSB of $L_i$ and $H_i$ according to Z, which results in mapping flag $U_i$. The following is the mapping rule equation:

$$U_i = Not\ (T_i \oplus C_{i\text{-MSB}})$$

The mapping rule of the embedding procedure is presented in Table 1.

**Table 1.** Mapping rule of embedding procedure.

| $T_i$ | MSB of $L_i$, $H_i$ | $U_i$ |
|---|---|---|
| 0 | 0 | 1 |
| 0 | 1 | 0 |
| 1 | 0 | 0 |
| 1 | 1 | 1 |

For example, assume that we have a cover picture I with a size of 8 × 8 pixels. First, divide the cover picture into 4 × 4 nonoverlapping segments, $C_i = 4$. Then, identify and convert the lowest and highest values, $L_i$ and $H_i$. The cover image preparation is presented in Figure 2. Therefore, on the basis of Equation (1), we can embed 8 bits of secret data. Suppose the characters of secret data are A and B, and the decimal values are 65 and 66, respectively. Convert them into seven-binary format $T_i$ = 1,0,0,0,0,0,1,1,0,0,0,0,1,0. The embeddable $T_i$ = 1,0,0,0,0,0,1,1. Suppose that mapping key Z = 7,8,1,2,3,4,6,5. Lastly, on the basis of the rule in Table 1, $U_i$ = 0 0 1 1 1 1 0 0, as shown in Figure 3. Figure 3 is a continuation of Figure 2, in which the secret bits are mapped to the lowest and highest segment values. The first secret bit is mapped with the 7th MSB, so that 1 is mapped with 0 resulting in 0. This is followed by the second secret bit, which is mapped with the 8th MSB, the third secret bit mapped with the 1st MSB, the fourth secret bit mapped with the 2nd MSB, the fifth secret bit mapped with the 3rd MSB, the sixth secret bit mapped with the 4th MSB, the seventh secret bit mapped with the 6th MSB, and the eight secret bit mapped with the 5th MSB.

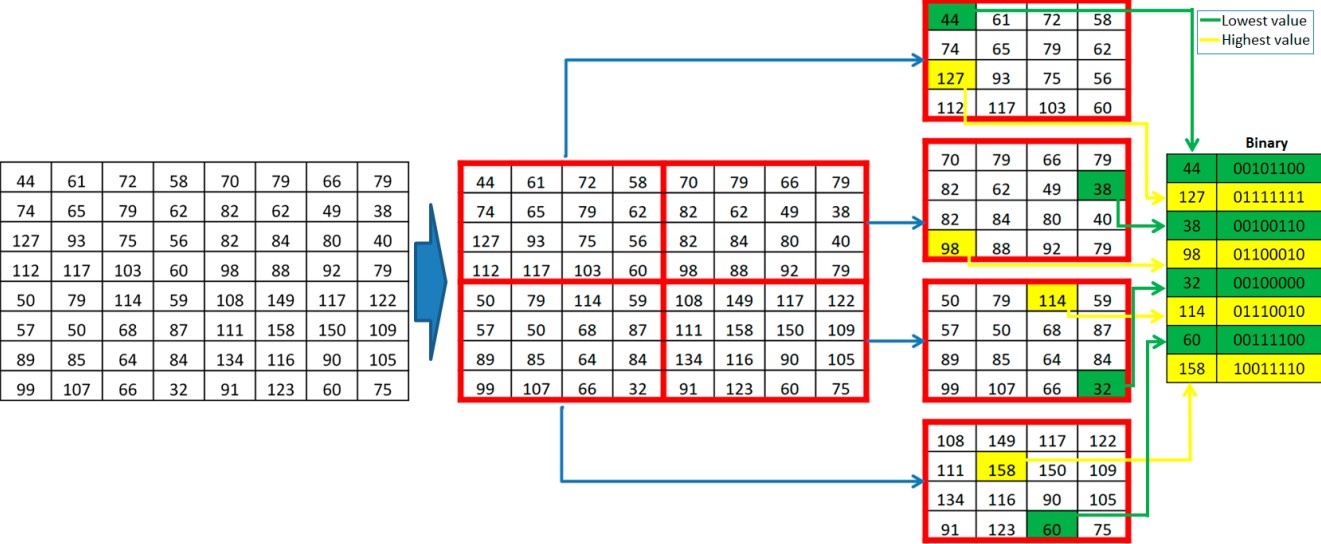

**Figure 2.** Example of the preparation of a cover image.

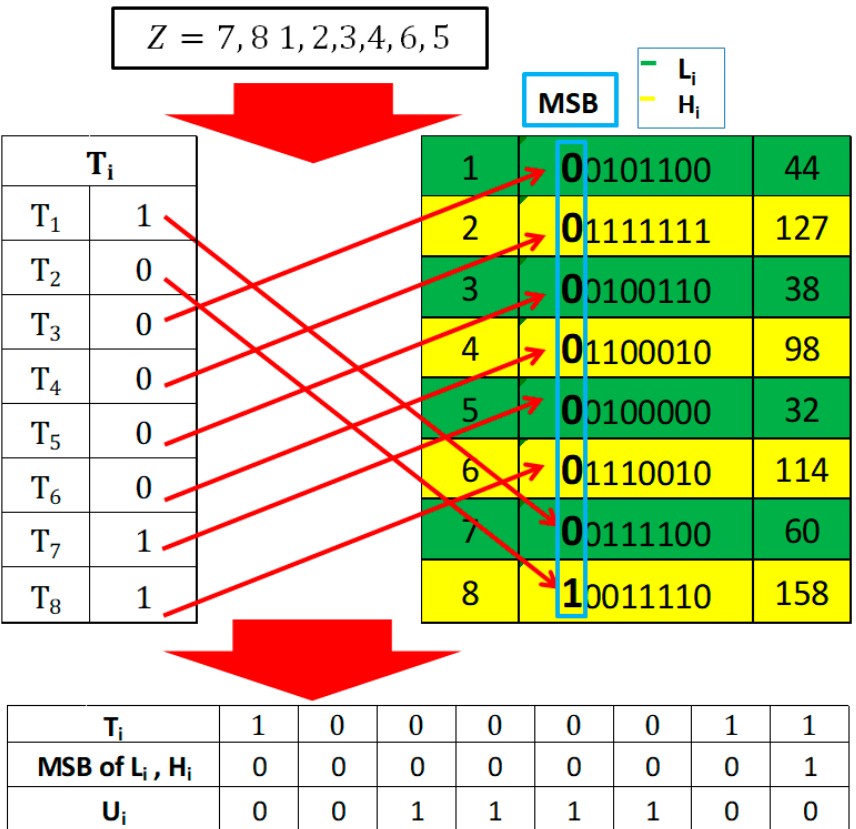

**Figure 3.** Example of mapping.

### 2.2. Extracting Procedures

The extracting procedures include two important processes: cover image preparation and mapping. The extraction technique is described in full below:

Stego Image Preparation

1. Divide a stego picture S with size J × K pixels into j × k nonoverlapping segments, $D_i$.
2. Determine the lowest and highest values, $L_i$ and $H_i$.
3. Convert $L_i$ and $H_i$ into an eight-bit binary.

Mapping

Construct a mapping between $U_i$ and the MSB of $L_i$, $H_i$ in accordance with Z, resulting in secret data $T_i$. The mapping rule of the extraction procedure is shown in Table 2.

**Table 2.** Mapping rule of extraction procedure.

| $U_i$ | MSB of $L_i$, $H_i$ | $T_i$ |
|---|---|---|
| 1 | 0 | 0 |
| 0 | 1 | 0 |
| 0 | 0 | 1 |
| 1 | 1 | 1 |

## 3. Experimental Results and Comparison

To evaluate the performance of the proposed CIHLHF, we compare it to that of CIHMSB on six test grayscale images, namely, Airplane, Baboon, Barbara, Boat, Lena, and Pepper, as shown in Figure 4a–f, respectively. To assess image quality, we employed several indicators, including peak signal-to-noise ratio (PSNR), structural similarity (SSIM), and Q.

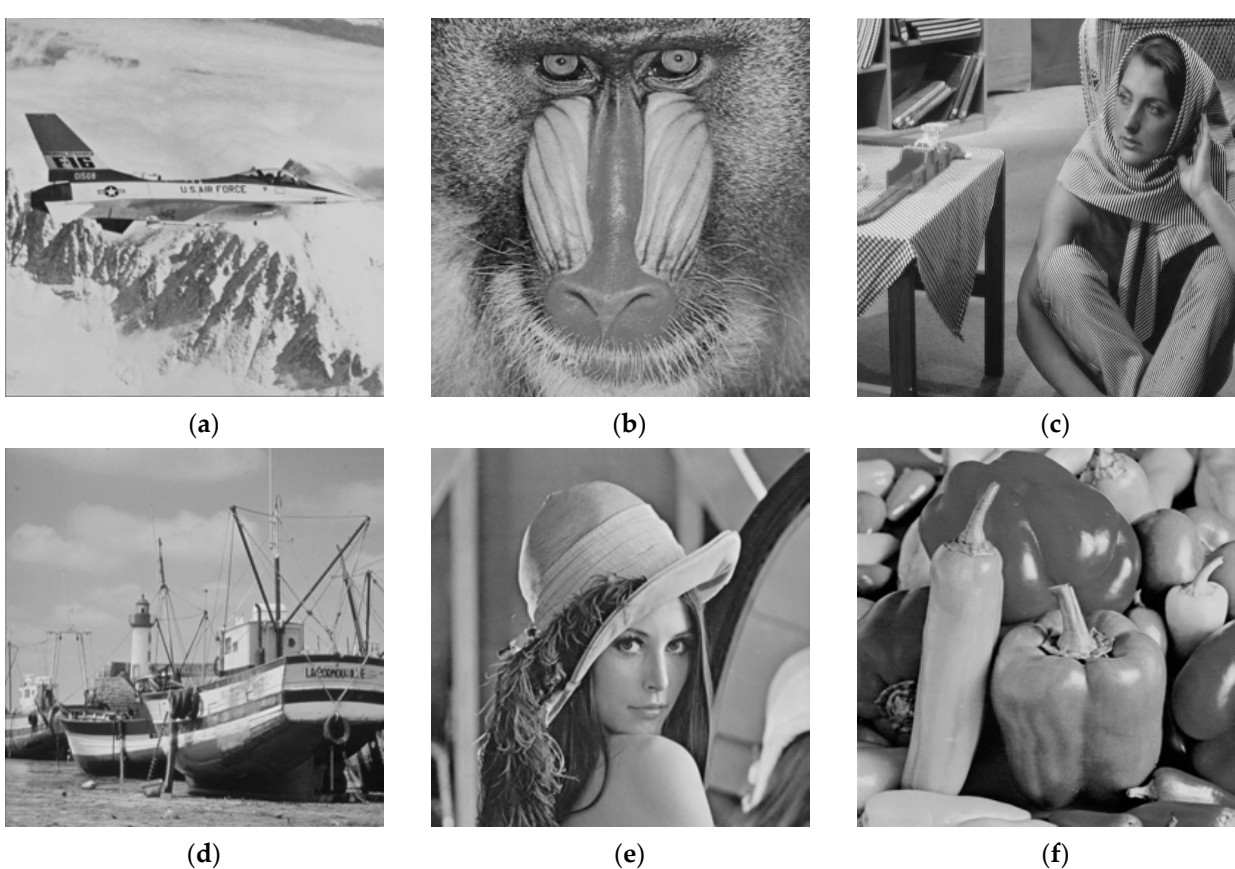

**Figure 4.** Grayscale test images: (**a**) Airplane; (**b**) Baboon; (**c**) Barbara; (**d**) Boat; (**e**) Lena; (**f**) Pepper.

PSNR is typically used to assess the quality between original and modified images after embedding (stego image). PSNR calculation is denoted in Equation (3), and the mean square error (MSE) is denoted in Equation (4):

$$\text{PSNR} = 10\log_{10}\frac{255^2}{\text{MSE}} \tag{3}$$

$$\text{MSE} = \frac{1}{M \times M}\sum_{i=1}^{M}\sum_{j=1}^{M}\left(L_{ij} - L'_{ij}\right)^2 \tag{4}$$

where $L_{ij}$ denotes the pixel location of image L located in the i-th row and the j-th column, and $L'_{ij}$ indicates a pixel position of stego image $L'$ located in the i-th row and the j-th column.

The structural similarity (SSIM) index is used to compare the similarity of a cover image and a stego image. It has a value between −1 and +1. When a cover image and a stego image are the same, SSIM equals to 1, which is also the optimal value of SSIM. It can be expressed as Equation (5):

$$\text{SSIM} = \frac{\left(2\overline{pq} + c_1\right)\left(2\sigma_{xy} + c_2\right)}{\left[(\overline{p})^2 + (\overline{q})^2 + c_1\right]\left(\sigma_x^2 + \sigma_y^2 + c_2\right)} \tag{5}$$

where $\overline{p}$ and $\overline{q}$ denote the average pixel values of the cover and stego images. $\sigma_x^2$ and $\sigma_y^2$ represent the standard deviation of the cover and stego images, while $\sigma_{xy}$ represents the covariance between the cover and stego images. Constant $c_1 = 2.5$, $c_2 = 7.65$.

Another important criterion for determining the similarity between cover and stego images is the universal image quality index ($Q_i$). When a cover image and a stego image

are exactly the same, $Q_i$ can obtain the optimal value of 1. The following is a definition of $Q_i$:

$$Q_i = \frac{4\sigma_{xy}\overline{pq}}{\left(\sigma_x^2 + \sigma_y^2\right)\left[(\overline{p})^2 + (\overline{q})^2\right]} \tag{6}$$

where $\overline{p}$ and $\overline{q}$ represent the average pixel values of the cover and stego images, respectively. $\sigma_x^2$ and $\sigma_y^2$ represent the standard deviation of the cover image and the stego image, while $\sigma_{xy}$ represents the covariance between the cover image and the stego image.

In addition, to evaluate the performance of CIHLHF, we compared it with that of other methods. In this study, we compared CIHLHF with CIHMSB [35]. The main reason is that CIHMSB provides a larger hiding capacity than that of the prior method in the same approach [31–34]. The main difference between the proposed scheme and CIHMSB is that, instead of mapping a secret message to the average value of MSB, the proposed scheme maps the secret message to the MSB's maximum and minimum.

Table 3 presents the performance comparison of CIHMSB and CIHLHF in terms of image quality (PSNR, SSIM, and $Q_i$) and hiding capacity. As shown in Table 3, the PSNR, SSIM, and $Q_i$ of the CIHMSB and CIHLHF techniques achieved optimal values of $\infty$, 1, and 1, respectively. The fundamental reason is that neither the CIHMSB nor the CIHLHF procedure modified the cover image, in accordance with the concept of coverless data hiding, in which the cover image is identical to the stego image. PSNR value $\infty$ indicates that the pixel values of the original and stego images were the same.

**Table 3.** Performance comparison of CIHMSB and CIHLHF.

| Methods | Bits | Carrirer$^{-1}$ | Hiding Capacity (Bits∗ Carrirer$^{-1}$) | PSNR (dB) | SSIM | $Q_i$ |
|---|---|---|---|---|---|---|
| CIHMSB [35] | 4 | $(512 \times 512)/(8 \times 8)$ | 16,384 | $\infty$ | 1 | 1 |
| CIHLHF | 12 | $= 4096$ | 49,152 (3 times higher) | $\infty$ | 1 | 1 |

Furthermore, CIHLHF surpassed CIHMSB in terms of hiding capacity. As seen in Table 3, for cover image size of $512 \times 12$ and fragment size of $8 \times 8$, the hiding capacity was 16,384 and 49,152 for CIHMSB and CIHLHF, respectively. The prominent reason is that CIHLHF examines the fragment's lowest and highest values, whereas CIHMSB uses the fragment's average value. Figure 5 presents the hiding capacity comparison of CIHMSB and CIHLHF. CIHLHF's hiding capacity was clearly three times higher than that of CIHMSB.

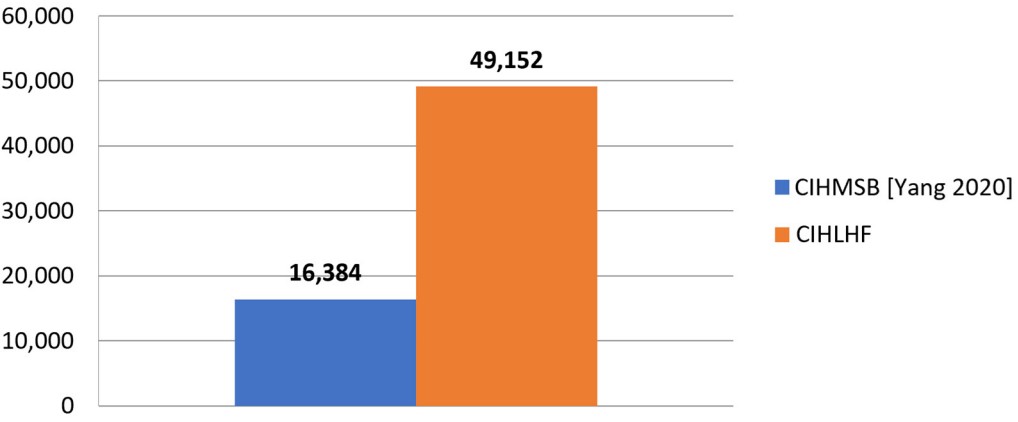

**Figure 5.** Hiding capacity graph (unit is (bits $\times$ carrirer$^{-1}$)) [35].

On the basis of the experimental results, we can conclude that the more pixel values used to embed the secret bits, the higher the hiding capacity is. So, we can generalize the hiding capacity of our proposed method as $k \times m$ bits, where $k = 1, 2, 3 \ldots n$ is the number of pixel values, and $m$ is the number of secret bits.

We analyzed the security of the CIHLHF method under the time cost of breaking the method with the scenario below. Assume that attackers have complete access to stego image S and mapping flag $U_i$. In this experiment, the cover image is a grayscale image with a size of $512 \times 512$. If the fragment size is $8 \times 8$, the number of segments $C_i$ is 4096.

For CIHLHF, for example, the number of secret bits $T_i$ is 49,152. So, when attackers need to extract 49,152 bits of secret data from the 4096th fragment, they would use the brute-force method because they have no information regarding mapping key Z. When the adversary needs to extract $T_i$ bits of secret information from $C_i$ image fragments, they must use brute-force attacks without knowing mapping key Z. Therefore, brute-force attacks can be calculated as follows:

$$U = \frac{(Ci)!}{(Ci - Ti)!} \tag{7}$$

Assuming that an ordinary computer can perform 10 billion calculations per second, it would take $2.69 \times 10^{177,113}$ years to extract the secret data.

Under the same scenario, CIHMSB would take $4.83 \times 10^{31,916}$ years to extract 16,384 bits from 4096 fragments. As seen in Table 4, with the same size of the cover image, due to the hiding capacity of CIHLHF being higher than that of CIHMSB, it took longer for the attack to extract secret data from CIHLHF than from CIHMSB. Therefore, CIHLHF is more secure than CIHMSB.

**Table 4.** Security comparison of CIHMSB and CIHLHF.

| Method | Total Bits | Time (Years) |
|---|---|---|
| CIHMSB [35] | 16,384 | $4.83 \times 10^{31,916}$ |
| CIHLHF | 49,152 | $2.69 \times 10^{177,113}$ |

Next, we discuss storage costs. Assume a cover image with size ($K \times L$) pixels and a segment block with size ($m \times n$) pixels. Thus, the number of fragments is m = [($K \times L$)/($m \times n$)]. In CIHMSB, the hiding capacity of the cover image is m, and the extra cost is needed to store mapping flag $U_i$ with m bits. In the proposed CIHLHF, the hiding capacity of the cover image is 2 m, and the extra cost is needed to store the mapping flag $U_i$ with 2 m bits. Although the mapping flag's size in the proposed CIHLHF is double that of CIHMSB, the hiding capacity in the proposed CIHLHF is double that of CIHMSB. In addition, the mapping flag's size is significantly smaller than that of the cover and stego images. Therefore, if we want to hide a secret message as with the proposed CIHLHF, 2 m bits, in CIHMSB, the size of a cover image needs to with 2($W \times H$) pixels, which is double the proposed CIHLHF size.

## 4. Conclusions

This work proposed high-capacity coverless image data hiding to solve the constraint of hiding capacity in coverless data hiding. Our proposed method exploits the lowest and highest pixels in each fragment. The secret data binary form is then mapped to the most significant bit of the lowest and highest binary forms. The mapping is ordered using predefined keys. The experimental findings suggest that our proposed technique has a higher hiding capacity than that of similar methods. Furthermore, our approach outperformed similar methods in terms of security.

**Author Contributions:** Conceptualization: K.A.; methodology: K.A.; software: S.-F.C.; validation: N.-I.W.; formal analysis: N.-I.W.; investigation: S.-F.C.; data curation: S.-F.C.; writing—original draft preparation: K.A.; writing—review and editing: M.-S.H.; project administration: M.-S.H.; funding acquisition: M.-S.H. All authors have read and agreed to the published version of the manuscript.

**Funding:** This research was funded by the Ministry of Science and Technology (Taiwan), grants number MOST 109-2221-E-468-011-MY3 and MOST 111-2622-8-468-001-TM1. The APC was funded by the Ministry of Science and Technology (Taiwan).

**Data Availability Statement:** Not applicable.

**Conflicts of Interest:** The authors declare no conflict of interest.

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
