# Peer review of "A High-Capacity Coverless Information Hiding Based on the Lowest and Highest Image Fragments"

_electronics, doi:10.3390/electronics12020395_

Round 1

Reviewer 1 Report

Overall contribution seems to be a minor change to the previous method, although the improvement seems significant. However, nothing comes for free, and it is unclear what the cost is to double the capacity. It is unfair to report increase in capacity without identifying such (hidden?) cost.

- please define all acronyms in Abstract when they are used first time, it is unclear why some terms are capitalized, other terms are not

- both in Abstract and the last but one paragraph in Introduction about the paper contributions are very brief which suggests that the change to the previous method is only minor; it would be very desirable to elaborate more on the paper contributions

- on p. 3, there is mixing of the same symbols with and without subscripts

- please better explain any trade-offs in improve the information hiding capacity, explain or comment on the values in Table 3, e.g. why PSNR is infinity (why is it even considered then?)

Author Response

The referees’ comments concerning our manuscript were extremely helpful in preparing a more explicit version. Therefore, we have revised our manuscript according to the referees’ and editor’s suggestions. We have marked the parts that have been changed in the revised manuscript or explained in red. The main revised parts are briefly described as attached file.

Reviewer 2 Report

1. Why use that it is chosen by the 44 127 38 98 in Fig. 2?

2. Why use that it is chosen by two same type 44 127 38 98 in Fig. 3?

3. Why use the Mapping Rule in Table 2?

4. Please show the SS in eq. (2).

5. Please show the equation of the hiding capacity.

6. Please show the equation of the Security comparison.

7.  It is too easy for the Mapping Rule.

 It is too easy for the Secret Data Preparation.

Round 2

Reviewer 1 Report

So if my understanding is right, the authors proposed to use twice as many bits to have twice the capacity. This looks like something very obvious. Would it be possible to generalize the proposed method to use k*m bits, k=1,2,3,....? Please consider adding this as a short subsection e.g. at the end of other results, even if the results are only presented in details for k=2.

L. 90: do not start a new paragraph.

Reviewer 2 Report

1. From Comment 2, the author's manuscript said "Figure 3 is a continuation of Figure 2, in which the secret bits are mapped to the lowest and highest segment values."

     I do not see any discussion from Reply 2.

  Please write in Reply 2.

2. From Comment 4, I do not see any SS = (S??? S???).

3. From Comment 5, the author's manuscript said "We have added a paragraph to explain it on Page 3."

    I do not see any discussion in Reply 5.

    Please write in Reply 5.

4. From Comment 6, the author's manuscript said "We have added some sentences to explain it on Page 8."

         I do not see any discussion in Reply 6.

        Please write in Reply 6.

5. From Comment 7, Sorry, it is too easy for the Mapping Rule.

    The teacher gives the example of Mapping Rule to teach the student.

6. From Comment 8, Sorry, it is too easy for the Secret Data Preparation.

The teacher gives the example of Secret Data Preparation to teach the student.

Round 3

Reviewer 2 Report

1. It is good to publish the Electronics.